# Efficacy and Accuracy of Maxillary Arch Expansion with Clear Aligner Treatment

**DOI:** 10.3390/ijerph20054634

**Published:** 2023-03-06

**Authors:** Gabriella Galluccio, Adriana A. De Stefano, Martina Horodynski, Alessandra Impellizzeri, Rosanna Guarnieri, Ersilia Barbato, Stefano Di Carlo, Francesca De Angelis

**Affiliations:** Department of Oral and Maxillofacial Sciences, Sapienza University of Rome, 00161 Rome, Italy

**Keywords:** maxillary expansion, clear aligner appliances, malocclusion, orthodontic appliances, digital orthodontics

## Abstract

The aim of this work was to evaluate the efficacy and accuracy of maxillary arch transverse expansion using the Invisalign^®^ clear aligner system without auxiliaries other than Invisalign attachments. Knowing the accuracy of a movement through a clear aligner system allows the clinician to plan the treatment with greater precision and to achieve the expected result faster. The study group included 28 patients with a mean age of 17 ± 3.2 years. The treatment protocol for all the selected patients included the application of the Invisalign^®^ clear aligner system without auxiliaries, except for the Invisalign^®^ attachments; in no case were tooth extraction or interproximal enamel reduction (IPR) performed. Linear measurements of the expansion were assessed before treatment (T0), at the end of treatment (T1), and on final virtual models by ClinCheck^®^ (TC). A paired *t*-test was used to compare T0-T1 and T1-TC differences. A paired *t*-test was applied, and one normality was validated with the Shapiro–Wilks test. If normality was not met, the nonparametric test (Mann–Whitney U test) was applied. The level of significance was set at 5%. Statistically significant differences were found for all measurements at T0-T1. The results showed an average accuracy of efficacy of 70.88%. The differences in predictability between the various vestibular measurements (intercanine, inter-premolar, and intermolar) were not statistically significant, while they were for gingival measurements. The overall accuracy of the expansion treatment was 70%, regardless of tooth type.

## 1. Introduction

The term “clear align therapy (CAT)” refers to the orthodontic technique with clear aligners for the treatment of dental malocclusions [1,2,3]. Since its development in 1997, Invisalign^®^ technology has been established worldwide as an aesthetic alternative to labial fixed appliances [1]. Since its first appearance on the market, the Invisalign^®^ system has seen significant development over time; many of its features have been continuously improved. New and different attachment designs have been developed, and the manufacturing material has been tested and improved. To allow for additional treatment biomechanics, the combined use of the clear aligner treatment with computer-guided piezocision and new auxiliaries, such as “precision cuts” and “Power Ridges”, has been proposed and used. According to the manufacturer, Invisalign^®^ is capable of effectively performing dental movements, such as bicuspid derotation, up to 50° and root movements of maxillary central incisors up to 4 mm. Despite the defended efficacy of the treatment, there is still controversy among professionals about the real clinical potency. On the one hand, the defenders are convinced and show cases of successful treatment, providing clinical evidence. In contrast, the opponents argue that there are significant limitations, especially when it comes to the treatment of cases with complex malocclusions [3,4,5,6]. Rossini et al., in their systematic literature review, found that the clear aligner treatment aligns and levels the arches and is effective in controlling anterior intrusion but not anterior extrusion. It is effective in controlling posterior buccolingual inclination but not anterior buccolingual inclination, and it is effective in controlling upper molar bodily movements of about 1.5 mm but is not effective in controlling the rotation of rounded teeth, in particular [5]. Aligners are now commonly used, such as in fixed appliance therapy, for the treatment of malocclusions of all types and severity, particularly for transverse dento-alveolar problems requiring the expansion of one or both arches [3,7].

In the evaluation of occlusion in the transverse plane, it is considered correct when the palatal cusp of the maxillary posterior teeth occludes with the central fossa of the mandibular posterior teeth [8,9]. If the upper buccal cusp occludes with the central fossa of the posterior lower teeth, a malocclusion occurs, which is called a crossbite [8]. This type of malocclusion may be of skeletal origin, whereby the dento-alveolar processes are correctly positioned in relation to the bony base, but the base presents maxillary skeletal hypoplasia or mandibular skeletal hyperplasia (or both) [8]. When the malocclusion is skeletal, its early correction is recommended through maxillary expansion with an orthopedic appliance, which guarantees greater stability over time [8]. When the malocclusion is of dental origin, the bone base has a correct transverse proportion, but dento-alveolar processes are altered [8,9,10]. It has been observed that one in three patients presents with a posterior crossbite of at least one tooth [10]. Arch expansion can be used to resolve crowding, correct dento-alveolar crossbite, or modify the arch shape [11]. Single-tooth crossbite is an easy case to treat with clear aligners; the aligners function as bite-planes that eliminate occlusal interferences and help to correct the crossbite. The crossbite of multiple teeth can be more complicated [9,10]. The aligners expand mainly by changing the torque of the posterior teeth through a crown buccal movement. The expansion can be performed at the canine, molar, and premolar level, or differentiated by maintaining a stable sector [7]. Several authors in their studies observed that treatment with the Invisalign^®^ system achieves a significant increase in the transverse measurements of the width of the arch as well as the perimeter of the arch [12,13,14]. Current knowledge on invisible aligners allows us to have a much clearer idea of the basic characteristics of aligner systems, but there remains a need to increase the use of systems other than Invisalign^®^ to provide greater evidence for different aligners that are widespread on the market [2].

The predictability of posterior expansion through treatment with aligners has been compared to the efficacy of the multibracket technique, and treatment with self-ligating multibrackets has been shown to be effective in solving mild crowding by increasing the width of the arch and correcting buccolingual tilt, occlusal contacts, and root angulations. While the Invisalign^®^ treatment aligns the arches by derotating the teeth and leveling the arches, due to the lack of control of tooth movement, Invisalign^®^ can easily tip crowns and be less effective in correcting transverse problems [15].

There is precedent in the literature for the effectiveness of Invisalign^®^ clear aligners (Align Technology, Santa Clara, CA, USA) and the predictability of its software (Align Technology, Santa Clara, CA, USA) for the planning of treatment with arch expansion. Some authors have evaluated how effective clear aligners are in achieving the proposed treatment objectives [14]; others have compared the results of treatment with clear aligners with those obtained with therapies using fixed appliances. Most of these investigations were carried out with the previous EX30 system, which was recently replaced by SmartTrack (Align Technology, Santa Clara, CA, USA), so it is necessary to evaluate the characteristics of the updated system. Posterior expansion of up to 2 mm per quadrant is a predictable movement achievable with aligners and decreases with increasing planned expansion [16,17]. It is advised, in case of crossbite, to overcorrect the expansion in the Clincheck^®^ programming until the palatal cusps of the upper molars contact the buccal cusps of the mandibular molars [18]. Beyond 2 mm of expansion, cross elastics or other auxiliaries may be necessary to achieve the planned result [16,17]. The predictability of maxillary expansion with clear aligners has shown wide variability over time.

Several studies that have evaluated the expansion of dental arches suggest that to minimize the risk of gingival recurrence and recession, the expansion limit of the arch width should be a maximum of 2–3 mm per quadrant. Invisalign^®^ may be indicated to achieve expansion in cases with crowding of 1 to 5 mm and in cases that require expansion to achieve space to include blocked out teeth. The expansion of the arch with Invisalign^®^ can result in an aesthetic advantage for the patient because, by widening the dental arches, it allows for improved aesthetics of the smile by reducing the buccal corridors [18,19,20,21].

Considering this variability in the results obtained from studies in the literature concerning the predictability of maxillary expansion with clear aligners, the aim of this study is to evaluate the efficacy and the accuracy of maxillary arch transverse expansion using the Invisalign^®^ clear aligner system without auxiliaries other than Invisalign^®^ attachments.

## 2. Materials and Methods

This prospective study was approved by the Ethical Committee of Sapienza University of Rome n° 1621/15 r. 3364, and the patients and/or their parents signed the informed consent for participation in the study.

The patients were selected from a group of 140 subjects recruited in the UOC of Orthodontics of the Department of Odontostomatological and Maxillo-Facial Science of “Sapienza” University of Rome. A total of twenty-eight patients were included in the study.

The patients were selected according to the following inclusion criteria: patients of both sexes, aged between 13 and 25 years old with complete permanent teeth, treatments performed with Invisalign^®^ aligners made from Smart-Track^®^ material, treatments that required transverse dento-alveolar expansion (2–4 mm) to correct malocclusion, patients with sufficient clinical crown height (greater than 4 mm), and patients who followed the treatment with good compliance. The exclusion criteria considered in the study were as follows: patients affected by systemic diseases and orofacial syndromes, patients with missing teeth in the posterior sectors, need for extractive therapy, presence of agenesis (excluding the third molar), excessive dental erosion at the cusp level such that the apex of the dental cusps cannot be found and multiple and/or advanced caries, patients with conoid teeth, patients with periodontal diseases, need for auxiliaries to correct transversal problems (TADs, REP, criss-cross elastics), patients with implants, prosthodontic rehabilitation or ankylosed teeth, and patients requiring orthognathic surgery.

All the patients were treated with the Invisalign^®^ technique by a single Invisalign provider. The treatment protocol for all the selected patients included the application of the Invisalign^®^ clear aligner system without auxiliaries except for the Invisalign^®^ attachments. In no cases were tooth extraction or interproximal enamel reduction (IPR) performed. Upper arch expansion was planned to correct crowding and transverse discrepancy. The patients were instructed on how to use the aligners: they should wear it all day, except during meals and dental hygiene, and all night; the change time between aligners was 7 days. The fit of the aligner and the presence of all attachments was checked by the provider every four stages. It was explained to all the patients that they were part of a research protocol and they or their parents accepted their participation by signing the informed consent; the patient’s collaboration was recorded in the clinical record.

For each patient, an intraoral scan of the pretreatment dental arches (T0) and a scan at the end of treatment (T1) were performed with the Itero Flex^®^ scanner. The final position of the corresponding ClinCheck^®^ representation (TC) was also collected to establish the accuracy of the final virtual model with respect to the movements observed in the post-treatment model.

Three models were then collected for each patient according to the following timetable:Pretreatment STL model (T0) obtained by scanning the maxillary arch before starting Invisalign^®^ treatment.Post-treatment STL model (T1) obtained from scanning the maxillary arch at the end of the treatment with Invisalign^®^.STL model from the final model programmed on the ClinCheck^®^ software (TC).

All models of the maxillary arches were opened with the program ExoCad^®^ (DentalCad). Using the program’s own measuring tool, linear millimeter measurements were taken. All measurements were performed by a trained single operator. The following transverse linear measurements were carried out on the upper arch for each T0 and T1 model and for the ClinCheck^®^ model (TC) (Figure 1):Intercanine cusp width: linear distance in millimeters between the cusp of the maxillary canine of one hemiarch to the cusp of the maxillary canine of the contralateral hemiarch (A).Intercanine gingival width: linear distance in millimeters between the most apical point of the palatal surface of the canine’s crown of the maxillary canine of one hemiarch to the same point of the contralateral hemiarch (B).First inter-premolar width: linear distance in millimeters between the buccal cusp of the first premolar of one hemiarch to the buccal cusp of the contralateral first premolar (C).Second inter-premolar width: linear distance in millimeters between the buccal cusp of the second premolar of one hemiarch to the buccal cusp of the contralateral first premolar (D).First molar mesio-vestibular cusp width: linear distance in millimeters between the mesiobuccal cusp of the first molar of one hemiarch to the mesiobuccal cusp of the contralateral first molar (E).First molar gingival width: linear distance in millimeters between the most apical point of the palatal surface of the first molar’s crown of one hemiarch to the same point of the contralateral hemiarch (F).

In addition, the following measurements were performed:Expansion obtained was calculated by the difference between the post-treatment distance with respect to the pretreatment amplitude (T1-T0).Planned expansion was calculated by the difference between the planned distance on the Clincheck^®^ with respect to the pretreatment amplitude (TC-T0).Accuracy of expansion was calculated by the difference between planned expansion on the Clincheck with respect to the obtained expansion (TC-T1).

Clinical accuracy (%) was achieved for all measurements, using the equation [(expansion obtained/planned expansion) × 100].

To estimate the size of the sample population for this study, a preliminary investigation was carried out to determine the power of the study (PS) and to establish the effect size (ES) (0,58) of the sampled population for the experimental study. Twenty-six patients were needed to estimate the expansion movement with a 95% confidence interval (CI), a power of 80%, and a level of significance of 5% for detecting an effect size of 0.58. Intra-examiner reliability was evaluated; the same examiner performed the measurements on 10 patients and repeated them two weeks later. The reliability of all measurements was assessed using an interclass correlation coefficient (ICC).

Numerical variables were expressed as mean and standard deviation values. Descriptive statistical analysis was performed for all measurements separately to compare the T0-T1 changes and the T0-TC differences. The normality of the measurements was assessed using the Shapiro–Wilks test. To compare the means between groups, a Student’s *t*-test was performed for independent data once normality was validated. If normality was not met, the nonparametric test (Mann–Whitney U test) was applied. The significance level applied in the analysis was 5% (α = 0.05). SPSS software (IBM Corp, Chicago, IL, USA) version 26 was used to analyze the data.

## 3. Results

The results obtained displayed a high degree of intra-observer reliability with an intraclass correlation coefficient > 0.80 for all linear measurements.

Twenty-eight patients (15 males, 18 females), with a mean age of 17 ± 3.2 years old were evaluated.

The Table 1 shows the descriptive statistics of all the measurements performed pretreatment (T0), post-treatment (T1), and in the Clincheck^®^ model (TC).

The planned expansion (TC-T0), the expansion obtained (T1-T0), the difference between expansion obtained and planned expansion, and the clinical accuracy are described in Table 2.

The planned expansion (mm) increased progressively from anterior to posterior at the level of the cusps, i.e., the planned intercanine width was on average smaller than the planned width of the first premolar, and the planned width of the first premolar was on average smaller than the planned width of the first molar. Furthermore, the planned expansions in millimeters for intercanine and intermolar gingival width were less than those for the cusp width.

On average, an expansion of between 5% and 7% more than the initial width (between 1.6 mm and 3.5 mm) was planned. The maximum expansion was planned at the level of the first inter-premolar width (7.35%, 2.95 mm) and the minimum at the intercanine cusp width (4.86%, 1.6 mm). On average, an expansion of between 3% and 7% more than the initial width was obtained. The maximum expansion was obtained at the first inter-premolar width level (6.87%, 2.7 mm) and the minimum at the first intermolar gingival width level (2.92%, 0.98 mm). The percentage of expansion obtained was less than the percentage of expansion planned in all measures. The T1-TC difference was less than 1 mm, except for the width of the intermolar buccal cusp that reaches it. The greatest differences between T1 and TC occurred at the level of the intermolar buccal cusp width (1.05 mm) and at the level of the gingival width (intercanine gingival width 0.98 mm and first intermolar gingival width 0.78 mm). However, in the intercanine, inter-premolar, and intermolar measurements at the level of the cusps, the differences between the expansion obtained and the planned expansion were not statistically significant, while they were statistically significant for gingival measurements (intercanine gingival width, intermolar gingival width). This result suggests that there is more vestibular tipping movement than body movement of the crowns at the level of the canine and of first molars.

The global clinical accuracy of the expansion treatment was 70.88%. The accuracy of the gingival measurements was low, around 50%, while for the measurements of the cusp width, the accuracy was between 70% and 82%. In the intercusp measurements, the expansion was more accurate for the first premolar (93.53%) and less for the first molar (70.55%).

## 4. Discussion

This study evaluated the possibility of effective transversal expansion of the upper arch through Invisalign^®^ treatment without the use of auxiliaries other than Invisalign^®^ attachments and the difference at different levels. In addition, the accuracy of the virtual pretreatment model developed with ClinCheck^®^ was evaluated in relation to the results obtained on from transversal expansion of the maxillary arch.

Monitoring tooth movement in orthodontics is important to assess the ability of devices to achieve movement and establish protocols capable of achieving orthodontic treatment goals [22,23,24,25]. New technologies facilitate the evaluation of dental movement and allow for more precise measurements [22,23,24,25]. In this way, it was possible to evaluate the possibility of expansion with Invisalign^®^. The results show that it is possible to expand to a higher percentage at the intercuspid level of the molar area and less at the canine intercuspid level. These results are in line with Morales-Burruezo et al. [18] who analyzed transverse expansion using Invisalign SmartTrack and concluded that expansion is achievable when it is alveolar, with higher efficiency at the premolar level and lower at the canine level. However, Clemens et al. [19], who evaluated using the Peer Assessment Rating index (PAR index) in 51 patients treated with aligners, observed that of the 25 patients who required transverse augmentation, 79% did so, resulting in 17% remaining stable and 4% worsening.

To assess the accuracy of expansion, an effectiveness index was considered, i.e., the closer the expansion obtained was to that predicted by the ClinCheck^®^. Effectiveness was considered to be 100% if the expansion obtained was statistically equal to that predicted. The results of this study showed an average accuracy of effectiveness of 70%. The differences in accuracy between the different measures (intercanine cusp and gingival width, first inter-premolar width and first intermolar cusp and gingival width) were not statistically significant; therefore, the overall accuracy of the expansion treatment was 70%, regardless of tooth type.

The present study showed that the effectiveness is lower when measured at the palatal side of the tooth, in agreement with Houle et al. [20], who claimed that body movement is not possible but instead a coronal inclination of the tooth. Furthermore, they state that the accuracy of digital programming with aligners is 72.8% in the maxillary arch, in accordance with our results. In our study, the effectiveness was on average 55% at the intermolar gingival level, while at the canine gingival level, it was 43%, and these results suggest, as reported in other studies [20,26,27], that there is less movement of the root portion of the tooth compared to the cusp portion, at least at the canine and molar levels. It would therefore appear that, although a body movement is programmed in the ClinCheck^®^, what is obtained is mainly a tipping coronal movement of the tooth.

Kraviz et al. [21] analyzed the predictability of Invisalign treatment with G3 material by superimposing initial and final models and showed that transverse expansion is not very accurate, with a predictability of 40.7%. The authors state that any type of movement has a predictability of 41%. However, it should be noted that the authors analyzed the effectiveness of the expansion with aligners made of G3 material, while the present study analyzed the results with the use of the new SmartTrack^®^ material. This difference could explain a better performance of the new material to which the expansive force is applied.

Similar studies were performed by Lione et al. [26,27] on the analysis of dental expansion movements in digital dental models. In agreement with the present study, they obtained a greater expansion at the level of the upper first molars with respect to other teeth. In their study, linear and angular measurements were performed before treatment (T0), at the end of treatment (T1), and on final virtual models (ClinCheck^®^ models), and significant differences were obtained for both linear and angular measurements for maxillary canines, resulting in little predictability [26].

In another study, Lione et al. [27] evaluated maxillary expansion with the Invisalign First System^®^ in growing subjects. Twenty-three patients with a mean age of 9.4 ± 1.2 years old, with a maxillary posterior transverse interarch discrepancy, were included in the study. The discrepancy was obtained by calculating the difference between the maxillary intermolar width, measured between the central fossae of the maxillary first molars on each side, and the mandibular intermolar width, measured between the mesiobuccal cusps of the mandibular first molars on each side. Patients were treated without extraction with Invisalign First System^®^ clear aligners with no auxiliaries other than Invisalign^®^ attachments, and no interproximal enamel reduction (IPR) was planned during treatment, as in our protocol. The results of their study showed a significant increase in the greatest width in the first primary molars compared to the second primary molars and primary canines. Maxillary first molars also showed the greatest expansion in mesial intermolar width due to rotation that occurred during expansion around the palatal root of the hinge tooth. These results are consistent with ours in that the greatest expansion was obtained in the most posterior sectors and at the occlusal level; however, in our study we did not consider both cusps of the molar, so it was not possible to assess whether rotation was present. This study has some limitations; for example, the amount of crowding that could influence the effectiveness of the expansion treatment was not considered, and the patients were not classified according to the amount of expansion needed considering the crowding. For future research, it would be advisable to increase the size of the sample, considering different groups of malocclusions and include a control group with another type of appliance useful for dento-alveolar expansion. In addition, other measures could be included to evaluate the vestibular inclination of the teeth and the rotation as a treatment effect to confirm the promising results of the present study.

## 5. Conclusions

Experience has shown us that certain movements cannot be achieved with aligners, but the actual limitations are unclear. Previsualization of the result can often be misleading for clinicians and patients. In conclusion, the efficacy in maxillary arch transverse expansion, on average, is rated at 70%, and is not related to the type of tooth considered but applies overall. Effectiveness is lower at the lingual level, with an average of 55% at the intermolar level, and 46% at the canine level. Statistically significant differences were found between the efficacy at the cuspal level compared to the efficacy measured at the most apical point of the palatal surface of the tooth, indicating that there is more tipping movement than body movement. The ClinCheck programs a body movement, whereas what we have obtained is a tipping movement.

## Figures and Tables

**Figure 1 ijerph-20-04634-f001:**
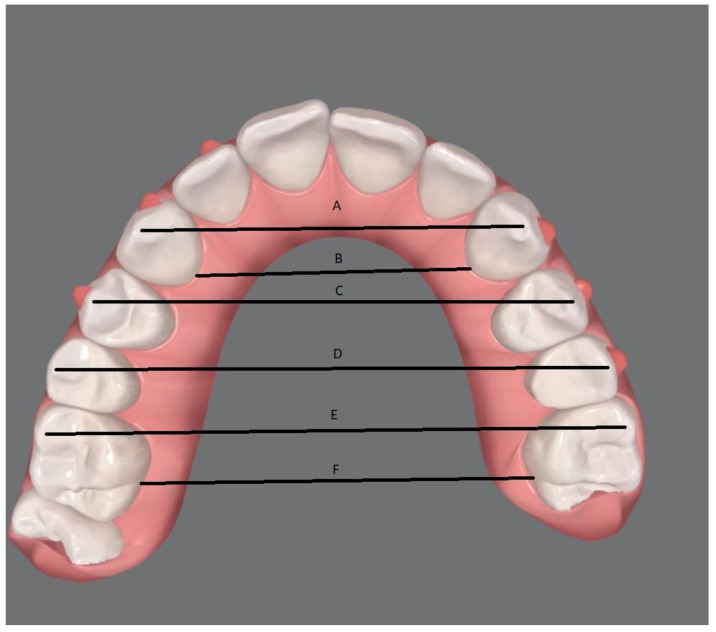
Transverse linear measurements performed on the maxillary arch for each model at T0, T1, and TC. (**A**) Intercanine cusp width; (**B**) intercanine gingival width; (**C**) first inter-premolar width; (**D**) second inter-premolar width; (**E**) first molar mesio-vestibular cusp width; (**F**) first molar gingival width.

**Table 1 ijerph-20-04634-t001:** Descriptive statistics of the measurements performed pretreatment (T0), post-treatment (T1), and in the Clincheck model (TC).

	Pretreatment (T0)		Post-Treatment (T1)		Clincheck (TC)	
Measurements	Mean	SD	Mean	SD	Mean	SD
Intercanine cusp width	33.52	1.93	34.85	1.63	35.15	1.78
Intercanine gingival width	23.73	2.20	24.50	1.75	25.48	1.33
First inter-premolar width	40.16	2.25	42.92	2.95	43.11	2.01
Second inter-premolar width	45.13	2.46	47.54	3.03	48.17	2.45
First intermolar cusp width	49.73	3.16	52.26	3.58	53.31	3.18
First intermolar gingival width	33.53	2.71	34.51	1.76	35.29	2.83

**Table 2 ijerph-20-04634-t002:** Difference between measurements in planned expansion (TC-T0) and the expansion obtained (T1-T0), and clinical accuracy (%).

	Planned Expansion TC-T0		Expansion Obtained T1-T0			Obtained-Planned T1-TC		Clinical Accuracy (%)
Measurements	mm	%	mm	*p*-Value	%	mm	*p*-Value	
Intercanine cusp width	1.63	4.86%	1.33	<0.05 *	3.97%	−0.29	>0.05	81.99%
Intercanine gingival width	1.75	7.37%	0.77	<0.05 *	3.24%	−0.98	<0.05 *	43.90%
First inter-premolar width	2.95	7.35%	2.76	<0.05 *	6.87%	−0.19	>0.05	93.53%
Second inter-premolar width	3.03	6.71%	2.41	<0.05 *	5.34%	−0.62	>0.05	79.43%
First intermolar cusp width	3.58	7.20%	2.52	<0.05 *	5.07%	−1.05	>0.05	70.55%
First intermolar gingival width	1.76	5.25%	0.98	<0.05 *	2.92%	−0.78	<0.05 *	55.85%
Mean	2.45	6.46%	1.80		4.57%	−0.65		70.88%

* = statistically significant.

## Data Availability

The data presented in this study are unavailable due to privacy or ethical restrictions.

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
