# Peer review of "Efficacy and Accuracy of Maxillary Arch Expansion with Clear Aligner Treatment"

_ijerph, 2023, doi:10.3390/ijerph20054634_

Round 1
Reviewer 1 Report
Dear Authors,
Thank you for the opportunity to review your manuscript. I offer the following recommendations for your consideration:
- State who give ethical clearance for the study, give the approval number.
- State what statistical methods you used to evaluate the data in abstract and materials and methods.
- Please show how to calculate sample size.
- How to appraise intra observer agreement?
- How to quantify the intra observer measurement error?
- Please give p-value in table 1,2.
- Authors should compare and contrast your results in each area (at intercanine cusp and gingiva width, interpremolar width, intermolar cusp and gingiva width) with other similar studies
and give the reasons why give the same or different results.
- The discussion section should have a summary; this is not the same as a conclusion.
- Please check reference no.9 that have 2 references.
Reviewer 2 Report
Dear authors!
The presented article is aimed at investigating the effectiveness and predictability of the results of maxillary arch expansion with clear aligner treatment, for which you have chosen a statistical method for analyzing the reliability of data differences. However, the use of a criterion for comparing two data sets can only indicate the presence or absence of differences and cannot give basis for conclusions about predictive power in quantitative terms. From my point of view, predictivity is related to probability and the result of such an assessment should be an indicator reflecting the probability of practical achievement of theoretical expected results. Despite the many similar studies where the term "predictivity" is used, I propose to revise the terminology in the presented article.
The presented design of the study allows you to answer the question of the effectiveness of treatment by comparing the values before (T0) and after (T1), as well as the accuracy (but not predictability) of the planning method (prognosis) when comparing the values of T1 and TC.
However, in my opinion, the most interesting data on the comparative analysis of T1–TC values are not presented in sufficient detail in the manuscript. The description in "Materials and Methods" section [149-153] contains information only about comparison of T0-T1 and T0-TC values. There is no information that a T1-TC comparison was intended.
It is also not entirely clear which hypothesis you wanted to test when assessing the reliability of the differences T0-TC [150]?
I am not sure that it is correct to give a promising value ("Clinical Predictability") for all 6 transversal linear measurements (A–F), since the points for measurements B and F are soft tissues, and other measurements were made with respect to hard tissues.
Direct scanning of soft tissues with an intraoral scanner is associated with biological and technical aspects. Biological measurement error is associated with dimensional variability of the gum in the area of measurement points due to possible inflammatory processes. The technical aspect of the intraoral scanners’ accuracy with respect to soft tissues is also a problem, since this method is the least accurate compared to the traditional obtaining of impressions or obtaining impressions using a laboratory scanner. Perhaps these aspects could partially explain why in your study the values of measurements B and F were almost two times less than the other measurements, and also significantly different within the framework of the T1-TC assessment.
Meanwhile, in the study you cited Morales-Burruezo I. et al. [18], in the "Discussion" section, the authors also raised the question of the correctness of soft tissue measurements, however, based on another technical reason directly related to the peculiarity of the Clincheck program.
The text fragment [197-199] goes beyond the scope of your study aim. This text is about the correctness of the intraoral scanners used for measuring the movement of teeth. This information is worth noting in the "Introduction" section. Also exclude literary references [23-25], since the purpose and conclusions of these studies do not correspond to the indicated aim.
Round 2
Reviewer 1 Report
Dear Authors,
Thank you for the opportunity to review your revised manuscript. I offer the following recommendation for your consideration:
Please state the sample size formula and give the confidence level, confidence interval.
Author Response
Dear reviewer 1,
Thank you for your comments and contributions that have significantly improved the article.
Following your recommendation we have changed the paragraph of the sample size as follows:
To estimate the size of the sample population for this study, a preliminary investigation was carried out to determine the power of the study (PS) and to establish the effect size (ES) (0,58) of the sampled population for the experimental study,. Twenty-six patients were needed to estimate expansion movement with a 95% confidence interval (CI), a power of 80% and a level of significance of 5%, for detecting an effect size of 0.58.
You can find the modification in lines 202-207 of the new revised document.
We appreciate your time and dedication.
kind regards,
Adriana A. De Stefano.
Corresponding author
Reviewer 2 Report
Dear authors!
Thank you very much for the high efforts you’ve made in revising your article. It has significantly improved manuscript.
Author Response
Dear reviewer 2,
We appreciate the time dedicated and all the contributions you have made to the article, helping to improve it notoriously.
kind regards
Adriana A. De Stefano
Corresponding author